# Scale Adaptive Blind Deblurring

**Haichao Zhang**
Duke University, NC
hczhang1@gmail.com

**Jianchao Yang**
Adobe Research, CA
jiayang@adobe.com

## Abstract

The presence of noise and small scale structures usually leads to large kernel estimation errors in blind image deblurring empirically, if not a total failure. We present a scale space perspective on blind deblurring algorithms, and introduce a cascaded scale space formulation for blind deblurring. This new formulation suggests a natural approach robust to noise and small scale structures through tying the estimation across multiple scales and balancing the contributions of different scales automatically by learning from data. The proposed formulation also allows to handle non-uniform blur with a straightforward extension. Experiments are conducted on both benchmark dataset and real-world images to validate the effectiveness of the proposed method. One surprising finding based on our approach is that blur kernel estimation is not necessarily best at the finest scale.

## 1 Introduction

Blind deconvolution is an important inverse problem that gains increasing attentions from various fields, such as neural signal analysis [3, 10] and computational imaging [6, 8]. Although some results obtained in this paper are applicable to more general bilinear estimation problems, we will use blind image deblurring as an example. Image blur is an undesirable degradation that often accompanies the image formation process due to factors such as camera shake. Blind image deblurring aims to recover a sharp image from only one blurry observed image. While significant progress has been made recently [6, 16, 14, 2, 22, 11], most of the existing blind deblurring methods do not work well in the presence of noise, leading to inaccurate blur kernel estimation, which is a problem that has been observed in several recent work [17, 26]. Figure 1 shows an example where the kernel recovery quality of previous methods degrades significantly even though only $5\%$ of Gaussian noise is added to the blurry input. Moreover, it has been empirically observed that even for noise-free images, image structures with scale smaller than that of the blur kernel are actually harmful for kernel estimation [22]. Therefore, various structure selection techniques, such as hard/hysteresis gradient thresholding [2, 16], selective edge map [22], and image decomposition [24] are incorporated into kernel estimation.

In this paper, we propose a novel formulation for blind deblurring, which explains the conventional empirical coarse-to-fine estimation scheme and reveals some novel perspectives. Our new formulation not only offers the ability to encompass the conventional multi-scale estimation scheme, but also offers the ability to achieve robust blind deblurring in a simple but principled way. Our model analysis leads to several interesting and perhaps surprising observations: ($i$) Blur kernel estimation is not necessarily best at the finest image scale and ($ii$) There is no universal single image scale that can be defined as *a priori* to maximize the performance of blind deblurring.

The remainder of the paper is structured as follows. In Section 2, we conduct an analysis to motivate our proposed scale-adaptive blind deblurring approach. Section 3 presents the proposed approach, including a generalization to noise-robust kernel estimation as well as non-uniform blur estimation. We discuss the relationship of the proposed method to several previous methods in Section 4. Ex-

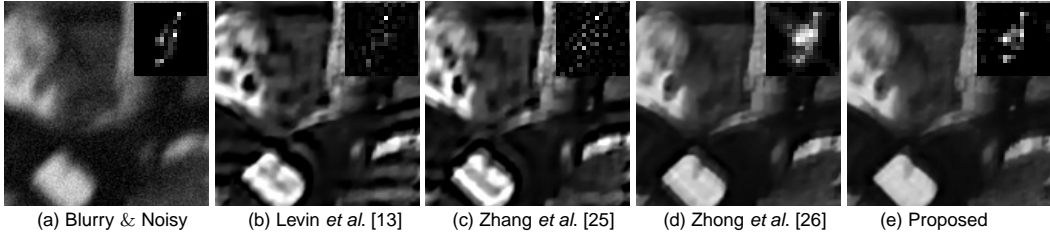

| (a) Blurry & Noisy | (b) Levin *et al.* [13] | (c) Zhang *et al.* [25] | (d) Zhong *et al.* [26] | (e) Proposed |
|---|---|---|---|---|

Figure 1: **Sensitivity of blind deblurring to image noise**. Random gaussian noise (5%) is added to the observed blurry image before kernel estimation. The deblurred images are obtained with the corresponding estimated blur kernels and the noise-free blurry image to capitalize the kernel estimation accuracy.

periments are carried out in Section 5, and the results are compared with those of the *state-of-the-art* methods in the literature. Finally, we conclude the paper in Section 6.

## 2   Motivational Analysis

For uniform blur, the blurry image can be modeled as follows

$$\mathbf{y} = \mathbf{k} * \mathbf{x} + \mathbf{n}, \tag{1}$$

where $*$ denotes 2D convolution,[1] $\mathbf{x}$ is the unknown sharp image, $\mathbf{y}$ is the observed blurry image, $\mathbf{k}$ is the unknown blur kernel (a.k.a., point spread function), and $\mathbf{n}$ is a zero-mean Gaussian noise term [6]. As mentioned above, most of the blind deblurring methods are sensitive to image noise and small scale structures [17, 26, 22]. Although these effects have been empirically observed [2, 22, 24, 17], we provide a complementary analysis in the following, which motivates our proposed approach later. Our analysis is based on the following result:

**Theorem 1 (Point Source Recovery [1])** *For a signal* $\mathbf{x}$ *containing point sources at different locations, if the minimum distance between sources is at least* $2/f_c$, *where* $f_c$ *denotes the cut-off frequency of the Gaussian kernel* $\mathbf{k}$, *then* $\mathbf{x}$ *can be recovered exactly given* $\mathbf{k}$ *and the observed signal* $\mathbf{y}$ *in the noiseless case.*

Although Theorem 1 is stated in the noiseless and non-blind case with a parametric Gaussian kernel, it is still enlightening for analyzing the general blind deblurring case we are interested in. As sparsity of the image is typically exploited in the image derivative domain for blind deblurring, Theorem 1 implies that large image structures whose gradients are distributed far from each other are likely to be recovered more accurately, which in return, benefits the kernel estimation. On the contrary, small image structures with gradients distributed near each other are likely to have larger recovery errors, and thus is harmful for kernel estimation. We refer these small image structures as *small scale structure* in this paper.

Apart from the above recoverability analysis, Theorem 1 also suggests a straightforward approach to deal with noise and small scale structures by performing blur kernel estimation after smoothing the noisy (and blurry) image $\mathbf{y}$ with a low-pass filter $\mathbf{f}_p$ with a proper cut-off frequency $f_c$

$$\mathbf{y}_p = \mathbf{f}_p * \mathbf{y} \Leftrightarrow \mathbf{y}_p = \mathbf{f}_p * \mathbf{k} * \mathbf{x} + \mathbf{f}_p * \mathbf{n} \Leftrightarrow \mathbf{y}_p = \mathbf{k}_p * \mathbf{x} + \mathbf{n}_p \tag{2}$$

where $\mathbf{k}_p \triangleq \mathbf{f}_p * \mathbf{k}$ and $\mathbf{n}_p \triangleq \mathbf{f}_p * \mathbf{n}$. As $\mathbf{f}_p$ is a low-pass filter, the noise level of $\mathbf{y}_p$ is reduced. Also, as the small scale structures correspond to signed spikes with small separation distance in the derivative domain, applying a local averaging will make them mostly canceled out [22], and therefore, noise and small scale structure can be effectively suppressed. However, applying the low-pass filter will also smooth the large image structures besides noise, and as a result, it will alter the profile of the edges. As the salient large scale edge structures are the crucial information for blur kernel estimation, the low-pass filtering may lead to inaccurate kernel estimation. This is the inherent limitation of linear filtering for blind deblurring. To achieve noise reduction while retaining the latent edge structures, one may resort to non-linear filtering schemes, such as anisotropic diffusion [20], Bilateral filtering [19], sparse regression [5]. These approaches typically assume the absence of motion blur, and thus can cause over-sharpening of the edge structures and over-smoothing of image details when blur is present [17], resulting in a filtered image that is no longer linear with respect to the latent sharp image, making accurate kernel estimation even more difficult.

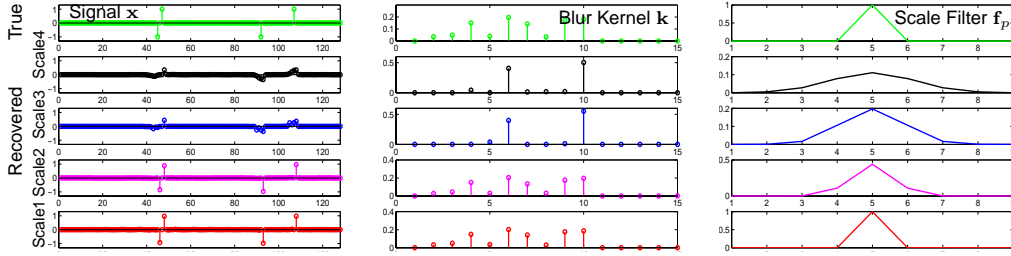

Figure 2: **Multi-Scale Blind Sparse Recovery**. The signal structures of different scales will be recovered at different scales. Large scale structures are recovered first and small structures are recovered later. Top: original signal, blur kernel. Bottom: the recovered signal and bluer kernel progressively across different scales (scale-4 to scale-1 represents the coarsest scale to the finest (original) scale. The blur kernel at the $i$-th scale is initialized with the solution from the $i$-1-th scale.

## 3 The Proposed Approach

To facilitate subsequent analysis, we first introduce the definition of *scale space* [15, 4]:

**Definition 1** *For an image* $\mathbf{x}$*, its scale-space representation corresponding to a Gaussian filter* $G_s$ *is defined by the convolution* $G_s * \mathbf{x}$*, where the variance $s$ is referred to as the scale parameter.*

Without loss of clarity, we also refer the different scale levels as different scale spaces in the sequel.

Natural images have a multi-scale property, meaning that different scale levels reveal different scales of image structures. According to Theorem 1, different scale spaces may play different roles for kernel estimation, due to the different recoverability of the signal components in the corresponding scale spaces. We propose a new framework for blind deblurring by introducing a variable scale filter, which defines the scale space where the blind estimation process is operated. With the scale filter, it is straightforward to come up with a blur estimation procedure similar to the conventional *coarse-to-fine* estimation by constructing an image pyramid. However, we operate deblurring in a space with the same spatial resolution as the original image rather than a downscaled space as conventionally done. Therefore, it avoids the additional estimation error caused by interpolation between spatial scales in the pyramid. To mitigate the problem of structure smoothing, we incorporate the knowledge about the filter into the deblurring model, which is different from the way of using filtering simply as a pre-processing step. More importantly, we can formulate the deblurring problem in multiple scale spaces in this way, and learn the contribution of each scale space adaptively for each input image.

### 3.1 Scale-Space Blind Deblurring Model

Our task is to recover $\mathbf{k}$ and $\mathbf{x}$ from the filtered observation $\mathbf{y}_p$, obtained via (2) with a known scale filter $\mathbf{f}_p$. The model is derived in the derivative domain, and we use $\mathbf{x} \in \mathbb{R}^m$ and $\mathbf{y}_p \in \mathbb{R}^n$ to denote the lexicographically ordered sharp and (filtered-) blurry image derivatives respectively.[2] The final deblurred image is recovered via a non-blind deblurring step with the estimated blur kernel [26]. From the modfied observation model (2), we can obtain the following likelihood:

$$p(\mathbf{y}_p|\mathbf{x}, \mathbf{k}, \lambda) \propto \exp\left[-\frac{\|\mathbf{f}_p * \mathbf{y} - \mathbf{f}_p * \mathbf{k} * \mathbf{x}\|_2^2}{2\lambda}\right] = \exp\left[-\frac{\|\mathbf{y}_p - \mathbf{k}_p * \mathbf{x}\|_2^2}{2\lambda}\right], \qquad (3)$$

where $\lambda$ is the variance of the Gaussian noise. Maximum likelihood estimation using (3) is ill-posed and further regularization over the unknowns is required. We use a parametrized Gaussian prior for $\mathbf{x}$, $p(\mathbf{x}) = \prod_i p(x_i) \propto \prod_i \mathcal{N}(x_i; 0, \gamma_i)$, where the unknown scale variables $\boldsymbol{\gamma} = [\gamma_1, \gamma_2, \cdots]$ are closely related to the sparsity of $\mathbf{x}$ and they will be estimated jointly with other variables. Rather than computing the Maximum A Posteriori (MAP) solution, which typically requires empirical tricks to achieve success [16, 2], we use type-II maximum likelihood estimation following [13, 21, 25], by marginalizing over the latent image and maximizing over the other unknowns

$$\max_{\boldsymbol{\gamma}, \mathbf{k}, \lambda \geq 0} \int p(\mathbf{y}_p|\mathbf{x}, \mathbf{k}, \lambda) p(\mathbf{x}) d\mathbf{x} \equiv \min_{\boldsymbol{\gamma}, \mathbf{k}, \lambda \geq 0} \mathbf{y}_p^T \boldsymbol{\Sigma}^T \mathbf{y}_p + \log|\boldsymbol{\Sigma}_p|, \qquad (4)$$

where $\boldsymbol{\Sigma}_p \triangleq \left(\lambda \mathbf{I} + \mathbf{H}_p \boldsymbol{\Gamma} \mathbf{H}_p^T\right)$, $\mathbf{H}_p$ is the convolution matrix of $\mathbf{k}_p$ and $\boldsymbol{\Gamma} \triangleq \mathrm{diag}[\boldsymbol{\gamma}]$. Using standard linear algebra techniques together with an upper-bound over $\boldsymbol{\Sigma}_p$,[3] we can reform (4) as follows [21]

$$\min_{\lambda, \mathbf{k} \geq 0, \mathbf{x}} \frac{1}{\lambda} \|\mathbf{f}_p * \mathbf{y} - \mathbf{f}_p * \mathbf{k} * \mathbf{x}\|_2^2 + r_p(\mathbf{x}, \mathbf{k}, \lambda) + (n - m) \log \lambda,$$

$$\text{with} \quad r_p(\mathbf{x}, \mathbf{k}, \lambda) \triangleq \sum_i \min_{\gamma_i} \frac{x_i^2}{\gamma_i} + \log(\lambda + \gamma_i \|\mathbf{k}_p\|_2^2), \tag{5}$$

which now resembles a typical regularized-regression formulation for blind deblurring when eliminating $\mathbf{f}_p$. The proposed objective function has one interesting property as stated in the following.

**Theorem 2 (Scale Space Blind Deblurring)** *Taking $\mathbf{f}_p$ as a Gaussian filter, solving (5) essentially achieves estimation for $\mathbf{x}$ and $\mathbf{k}$ in the scale space defined by $\mathbf{f}_p$ given $\mathbf{y}$ in the original space.*

In essence, Theorem 2 reveals the equivalence between performing blind deblurring on $\mathbf{y}$ directly while constraining $\mathbf{x}$ and $\mathbf{k}$ in a certain scale space and by solving the proposed model (5) with the aid of the additional filter $\mathbf{f}_p$. This places the proposed model (5) on a sound theoretical footing.

**Cascaded Scale-Space Blind Deblurring.** If the blur kernel $\mathbf{k}$ has a clear cut-off frequency and the target signal contains structures at distinct scales, then we can suppress the structures with scale smaller than $\mathbf{k}$ using a properly designed scale filter $\mathbf{f}_p$ according to Theorem 1, and then solve (5) for kernel estimation. However, in practice, the blur kernels are typically non-parametric and with complex forms, therefore do not have a clear cut-off frequency. Moreover, natural images have a multi-scale property, meaning different scale spaces reveal different image structures. All these facts suggests that it is not easy to select a fixed scale filter $\mathbf{f}_p$ a *priori* and calls for a *variable scale filter*.

Nevertheless, based on the basic point that large scale structures are more advantageous than small scale structures for kernel estimation, a natural idea is to perform (5) separately at different scales, and pick the best estimation as the output. While this is an appealing idea, it is not applicable in practice due to the non-availability of the ground-truth, which is required for evaluating the estimation quality. A more practical approach is to perform (5) in a cascaded way, starting the estimation from a large scale and then reducing the scale for the next cascade. The kernel estimation from the previous scale is used as the starting point for the next one. With this scheme, the blur kernel is refined along with the resolution of the scale space, and may become accurate enough before reaching the finest resolution level, as shown in Figure 2 for a 1D example. The latent sparse signal in this example contains $4$ point sources, with the minimum separation distance of $2$, which is smaller than the support of the blur kernel. It is observed that some large elements of the blur kernel are recovered first and then the smaller ones appear later at a smaller scale. It can also be noticed that the kernel estimation is already fairly accurate before reaching the finest scale (i.e., the original pixel-level representation). In this case, the final estimation at the last scale is fairly stable given the initialization from the last scale. However, performing blind deblurring by solving (5) in the last original scale directly (i.e., $\mathbf{f}_p \equiv \delta$) cannot achieve successful kernel estimation (results not shown).

A similar strategy by *constructing an image pyramid* has been applied successfully in many of the recent deblurring methods [6, 16, 2, 22, 8, 25]. It is important to emphasize that the main purpose of our scale-space perspective is more to provide complementary analysis and understanding of the empirical coarse-to-fine approach in blind deblurring algorithms, than to replace it. More discussions on this point are provided in Section 4. Nevertheless, the proposed alternative approach can achieve performance on par with state-of-the-art methods, as shown in Figure 4. More importantly, this alternative formulation offers us a number of extra dimensions for generalization, such as extensions to noise robust kernel estimation and scale-adaptive estimation, as shown in the next section.

### 3.2 Scale-Adaptive Deblurring via Tied Scale-Space Estimation

In the above cascade procedure, a single filter $\mathbf{f}_p$ is used at each step in a greedy way. Instead, we can define a set of scale filters $\mathcal{P} \triangleq \{\mathbf{f}_p\}_{p=1}^P$, apply each of them to the observed image $\mathbf{y}$ to get a set of filtered observations $\{\mathbf{y}_p\}_{p=1}^P$, and then *tie the estimation across all scales* with the shared latent sharp image $\mathbf{x}$. By constructing $\mathcal{P}$ as a set of Gaussian filters with decreasing radius, it is equivalent to perform blind deblurring in different scale spaces. Large scale space is more robust to image noise, and thus is more effective in stabilizing the estimation; however, only large scale

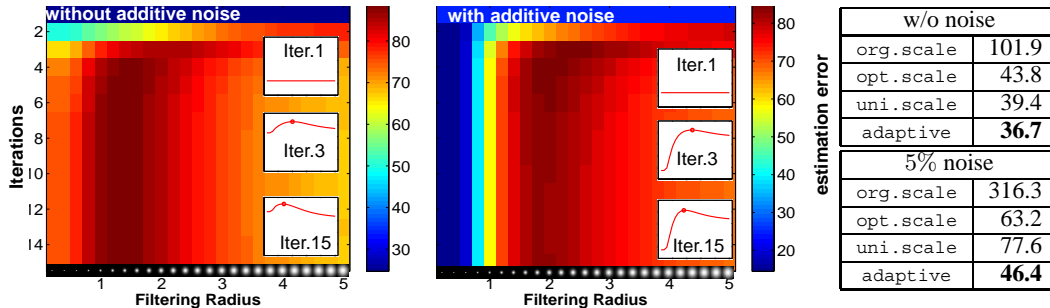

Figure 3: **Scale Adaptive Contribution Learning** for a set of 25 Gaussian filters with radius $r \in (0, 5]$ on the first image [14]. Left: without adding noise. Right: with $5\%$ additive noise. The values in the heat-map represent the contribution weight $(\lambda_p^{-1})$ for each scale filter during the iterations. The table on the right shows the performance (SSD error) of blind deblurring with different scales: original scale (`org.scale`), empirically optimal scale (`opt.scale`), multiple scales with uniform contribution weights (`uni.scale`) and multiple scales with adaptive weights (`adaptive`).

structures are "visible" (recoverable) in this space. Small scale space offers the potential to recover more fine details, but is less robust to image noise. By conducting deblurring in multiple scale spaces simultaneously, we can exploit the complementary property of different scales for robust blind deblurring in a unified framework. Furthermore, different scales may contribute differently to the kernel estimation, we therefore use a distinct noise level parameter $\lambda_p$ for each scale, which reflects the relative contribution of that scale to the estimation. Concretely, the final cost function can be obtained by accumulating the cost function (5) over all the $P$ filtered observations with adaptive noise parameters [4]

$$\min_{\{\lambda_p\}, \mathbf{k} \geq 0, \mathbf{x}} \sum_{p=1}^{P} \frac{1}{\lambda_p} \|\mathbf{f}_p * \mathbf{y} - \mathbf{f}_p * \mathbf{k} * \mathbf{x}\|_2^2 + R(\mathbf{x}, \mathbf{k}, \{\lambda_p\}) + (n-m) \sum_p \log \lambda_p,$$

$$\text{where} \quad R(\mathbf{x}, \mathbf{k}, \{\lambda_p\}) = \sum_p r_p(\mathbf{x}, \mathbf{k}, \{\lambda_p\}) = \sum_{p,i} \min_{\gamma_i} \frac{x_i^2}{\gamma_i} + \log(\lambda_p + \gamma_i \|\mathbf{k}_p\|_2^2).$$

(6)

The penalty function $R$ here is in effect a penalty term that exploits multi-scale regularity/consistency of the solution space. The effectiveness of the proposed approach compared to other methods is illustrated in Figure 1 and more results are provided in Section 5. Formulating the deblurring problem as (6), our joint estimation framework enjoys a number of features that are particularly appropriate for the purpose of blind deblurring in presence of noise and small scale image structures: $(i)$ It exploits both the regularization of sharing the latent sharp image $\mathbf{x}$ across all filtered observations and the knowledge about the set of filters $\{\mathbf{f}_p\}$. In this way, $\mathbf{k}$ is recovered directly without post-processing as previous work [26]; $(ii)$ the proposed approach can be extended to handle non-uniform blur, as discussed in Section 3.3; and $(iii)$ there is no inherent limitations on the form of the filters we can use besides Gaussian filters, e.g., we can also use directional filters as in [26].

**Scale Adaptiveness.** With this cost function, the contribution of each filtered observation $\mathbf{y}_p$ constructed by $\mathbf{f}_p$ is reflected by weight $\lambda_p^{-1}$. The parameters $\{\lambda_p^{-1}\}$ are initialized uniformly across all filters and are then learned during the kernel estimation process automatically. In this scenario, a smaller noise level estimation indicates a larger contribution in estimation. It is natural to expect that the distribution of the contribution weights for the same set of filters will change under different input noise levels, as shown in Figure 3. From the figure, we obtain a number of interest observations:

●The proposed algorithm is adaptive to observations with different noise levels. As we can see, filters with smaller radius contribute more in the noise-free case, while in the noisy case, filters with larger radius contribute more.

●The distribution of the contribution weights evolves during the iterative estimation process. For example in the noise-less case, starting with uniform weights, the middle-scale filters contribute the most at the beginning of the iterations, while smaller-scale filters contribute more to the estimation later on, a natural coarse-to-fine behavior. Similar trends can also be observed for the noisy case.

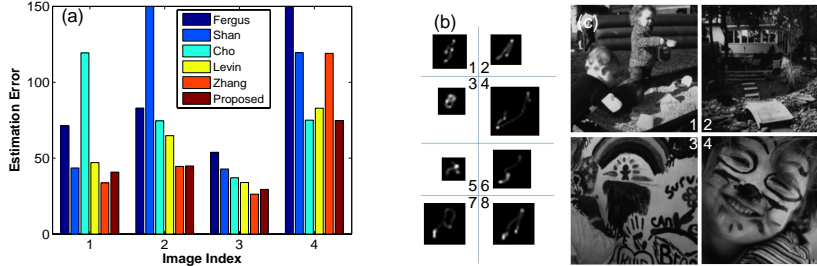

Figure 4: **Blind Deblurring Results: Noise-free Case.** (a) Performance comparison (image estimation error) on the benchmark dataset [14], which contains (b) 8 blur kernels and (c) 4 images.

- While it is expected that the original scale space is not the "optimal" scale for kernel estimation in presence of noise, it is somewhat surprising to find that this is also the case for the noise-free case. This corroborates previous findings that small scale structures are harmful to kernel estimation [22], and our algorithm automatically learn the scale space to suppress the effects of small scale structures.

- The weight distribution is more flat in the noise-free case, while it is more peaky for the noisy case. Figure 3 is obtained with the first kernel and image in Figure 4. Similar properties can be observed for different images/blurs, although the position of the empirical mode are unlikely to be the same.

The table in Figure 3 shows the estimation error using difference scale space configurations. Blind deblurring in the original space directly (`org.scale`) fails, indicated by the large estimation error. However, when setting the filter as $\mathbf{f}_o$, whose contribution $\lambda_o^{-1}$ is empirically the largest among all filters (`opt.scale`), the performance is much better than in the original scale directly, with the estimation error reduced significantly. The proposed method, by tying multiple scales together and learning adaptive contribution weights (`adaptive`), performs the best across all the configurations, especially in the noisy case.

### 3.3 Non-Uniform Blur Extension

The extension of the uniform blind deblurring model proposed above to the non-uniform blur case is achieved by using a generalized observation model [18, 9], representing the blurry image as the summation of differently transformed versions of the latent sharp image $\mathbf{y} = \mathbf{H}\mathbf{x} + \mathbf{n} = \sum_{j=1} w_j \mathbf{P}_j \mathbf{x} + \mathbf{n} = \mathbf{D}\mathbf{w} + \mathbf{n}$. Here $\mathbf{P}_j$ is the $j$-th projection or homography operator (a combination of rotations and translations) and $w_j$ is the corresponding combination weight representing the proportion of time spent at that particular camera pose during exposure. $\mathbf{D} = [\mathbf{P}_1\mathbf{x}, \mathbf{P}_2\mathbf{x}, \cdots, \mathbf{P}_j\mathbf{x}, \cdots]$ denotes the dictionary constructed by projectively transforming $\mathbf{x}$ using a set of transformation operators. $\mathbf{w} \triangleq [w_1, w_2, \cdots]^T$ denotes the combination weights of the blurry image over the dictionary. The uniform convolutional model (1) can be obtained by restricting $\{\mathbf{P}_j\}$ to be translations only. With derivations similar to those in Section 3.1, it can be shown that the cost function for the general non-uniform blur case is

$$\min_{\lambda, \mathbf{w} \geq 0, \mathbf{x}} \sum_{p=1}^{P} \frac{1}{\lambda_p} \|\mathbf{y}_p - \mathbf{H}_p \mathbf{x}\|_2^2 + \sum_{p,i} \min_{\gamma_i} \frac{x_i^2}{\gamma_i} + \log(\lambda_p + \gamma_i \|\mathbf{h}_{ip}\|_2^2) + (n - m) \sum_p \log \lambda_p, \quad (7)$$

where $\mathbf{H}_p \triangleq \mathbf{F}_p \sum_j w_j \mathbf{P}_j$ is the compound operator incorporating both the additional filter and the non-uniform blur. $\mathbf{F}_p$ is the convolutional matrix form of $\mathbf{f}_p$ and $\mathbf{h}_{ip}$ denotes the effective compound local kernel at site $i$ in the image plane constructed with $\mathbf{w}$ and the set of transformation operators.

## 4 Discussions

We discuss the relationship of the proposed approach with several recent methods to help understanding properties of our approach further.

**Image Pyramid based Blur Kernel Estimation.** Since the blind deblurring work of Fergus *et al.* [6], image pyramid has been widely used as a standard architecture for blind deblurring [16, 2, 8, 22, 13, 25]. The image pyramid is constructed by resizing the observed image with a fixed ratio for multiple times until reaching a scale where the corresponding kernel is very small, e.g. $3 \times 3$. Then the blur kernel is estimated firstly from the smallest image and is upscaled for initializing the next level. This process is repeated until the last level is reached. While it is effective for exploiting the

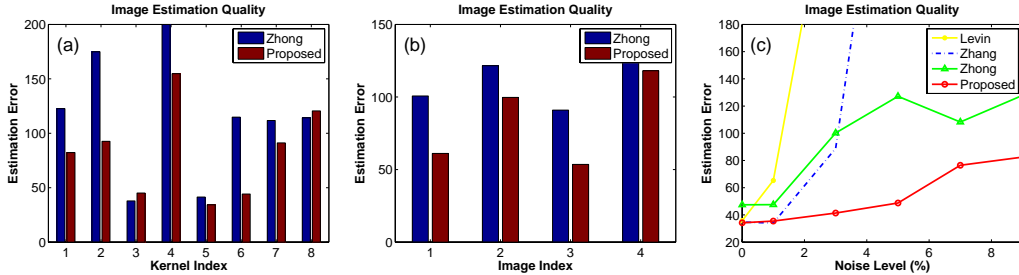

Figure 5: **Deblurring results in the presence of noise** on the benchmark dataset [14]. Performance averaged over (a) different images and (b) different kernels, with $5\%$ additive Gaussian noise. (c) Comparison of the proposed method with Levin *et al.* [13], Zhang *et al.* [25], Zhong *et al.* [26] on the first image with the first kernel, under different noise levels.

solution space, this greedy pyramid construction does not provide an effective way to handle image noise. Our formulation not only retains properties similar to the pyramid coarse-to-fine estimation, but also offers the extra flexibility to achieve scale-adaptive estimation, which is robust to noise and small scale structures.

**Noise-Robust Blind Deblurring [17, 26].** Based on the observation that using denoising as a preprocessing can help with blur kernel estimation in the presence of noise, Tai *et al.* [17] proposed to perform denoising and kernel estimation alternatively, by incorporating an additional image penalty function designed specially taking the blur kernel into account [17]. This approach uses separate penalty terms and introduces additional balancing parameters. Our proposed model, on the contrary, has a coupled penalty function and learns the balancing parameters from the data. Moreover, the proposed model can be generalized to non-uniform blur in a straightforward way. Another recent method [26] performs blind kernel estimation on images filtered with different directional filters separately and then reconstructs the final kernel in a second step via inverse Radon transform [26]. This approach is only applicable to uniform blur and directional auxiliary filters. Moreover, it treats each filtered observation independently thus may introduce additional errors in the second kernel reconstruction step, due to factors such as mis-alignment between the estimated compound kernels.

**Small Scale Structures in Blur Kernel Estimation [22, 2].** Based on the observation that small scale structures are harmful for kernel estimation, Xu and Jia [22] designed an empirical approach for structure selection based on gradient magnitudes. Structure selection has also been incorporated into blind deblurring in various forms before, such as gradient thresholding [2, 16]. However, it is hard to determine a universal threshold for different images and kernels. Other techniques such as image decomposition has also been incorporated [24], where the observed blurry image is decomposed into structure and texture layers. However, standard image decomposition techniques do not consider image blur, thus might not work well in the presence of blur. Another issue for this approach is again the selection of the parameter for separating texture from structure, which is image dependent in general. The proposed method achieves robustness to small scale structures by optimizing the scale contribution weights jointly with blind deblurring, in an image adaptive way.

The optimization techniques used in this paper has been used before for image deblurring [13, 21, 25], with different context and motivations.

## 5   Experimental Results

We perform extensive experiments in this section to evaluate the performance of the proposed method compared with several state-of-the-art blind deblurring methods, including two recent noise robust deblurring methods of Tai *et al.* [17], and Zhong *et al.* [26], as well as a non-uniform deblurring method of Xu *et al.* [23]. We construct $\{\mathbf{f}_p\}$ as Gaussian filters, with the radius uniformly sampled over a specified range, which is typically set as $[0.1, 3]$ in the experiment.[5] The number of iterations is used as the stopping criteria and is fixed as 15 in practice.

**Evaluation using the Benchmark Dataset of Levin *et al.* [14].** We first perform evaluation on the benchmark dataset of Levin *et al.* [14], containing $4$ images and $8$ blur kernels, leading to $32$ blurry images in total (see Figure 4). Performances for the noise-free case are reported in Figure 4, where the proposed approach performs on par with state-of-the-art. To evaluate the performances

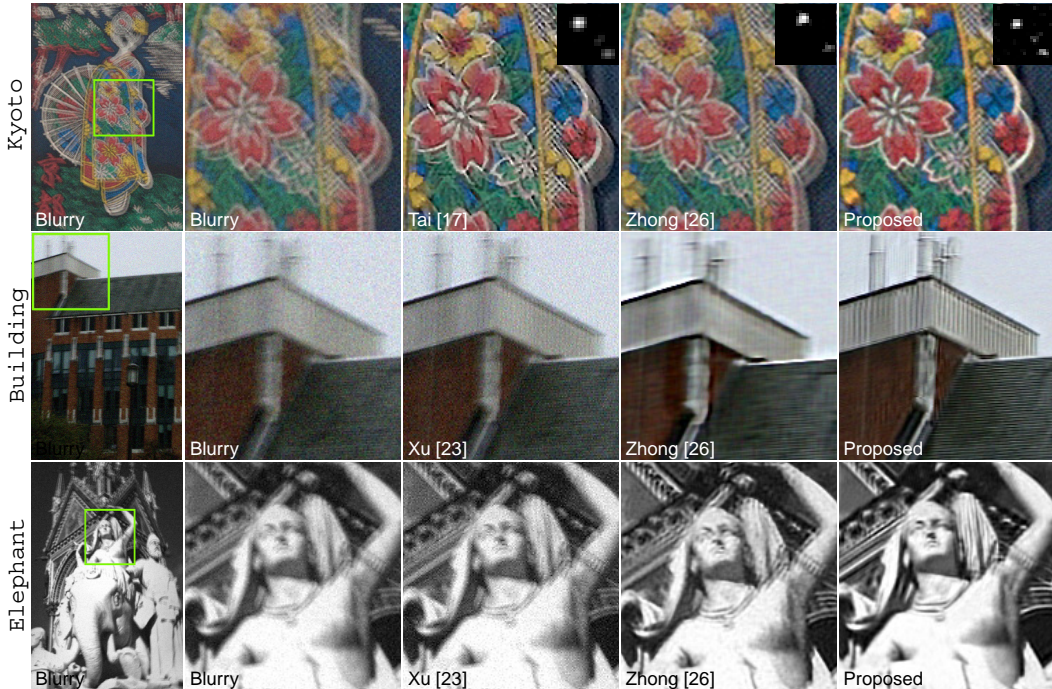

Figure 6: **Deblurring results** on image with non-uniform blur, compared with Tai *et al.* [17], Zhong *et al.* [26] and Xu *et al.* [23]. Full images are shown in the supplementary file.

of different methods in the presence of noise, we add i.i.d. Gaussian noise to the blurry images, and then perform kernel estimation. The estimated kernels are used for non-blind deblurring [12] on the noise-free blurry images. The bar plots in Figure 5 show the sum-of-squared-difference (SSD) error of the deblurred images using the proposed method and the method of Zhong *et al.* [26] when the noise level is 5%. As the same non-blind deblurring method is used, this SSD error reflects the quality of the kernel estimation. It is clear that the proposed method performs better than the method of Zhong *et al.* [26] overall. We also show the results of different methods with increasing noise levels in Figure 5. It is observed that while the conventional methods (e.g. Levin *et al.* [13], Zhang *et al.* [25]) performs well when the noise level is low, their performances degrade rapidly when the noise level increases. The method of Zhong *et al.* [26] performs more robustly across different noise levels, but does not performs as well as the other methods when the noise level is very low. This might be caused by the loss of information during its two-step process. The proposed method outperforms the other methods for all the noise levels, proving its effectiveness.

**Deblurring on Real-World Images.** We further evaluate the performance of the proposed method on real-world images from the literature [17, 7, 8]. The results are shown in Figure 6. For the Kyoto image from [17], the deblurred image of Tai *et al.* [17] has some ringing artifacts while the result of Zhong *et al.* [26] has ghosting effects due to the inaccurate kernel estimation. The deblurred image from the propose method has neither ghosting or strong ringing artifacts. For the other two test images, the non-uniform deblurring method [23] produces deblurred images that are still very blurry, as it achieves kernel estimations close to a delta kernel for both images, due to the presence of noise. The method of Zhong *et al.* [26] can only handle uniform blur and the deblurred images have strong ringing artifacts. The proposed method can estimate the non-uniform blur accurately and can produce high-quality deblurring results better than the other methods.

# 6 Conclusion

We present an analysis of blind deblurring approach from the scale-space perspective. The novel analysis not only helps in understanding several empirical techniques widely used in the blind deblurring literature, but also inspires new extensions. Extensive experiments on benchmark dataset as well as real-world images verify the effectiveness of the proposed method. For future work, we would like to investigate the extension of the proposed approach in several directions, such as blind image denoising and multi-scale dictionary learning. The task of learning the auxiliary filters in a blur and image adaptive fashion is another interesting future research direction.

**Acknowledgement** The research was supported in part by Adobe Systems.

## Footnotes

[1] We also overload $*$ to denote the 2D convolution followed by lexicographic ordering based on the context.

[2]The derivative filters used in this work are $\{[-1, 1], [-1, 1]^T\}$.

[3]$\log |\boldsymbol{\Sigma}_p| \leq \sum_i \log \left(\lambda + \gamma_i \|\mathbf{k}_p\|_2^2\right) + (n - m) \log \lambda$ [25].

[4]This can be achieved either in an online fashion or in one shot.

[5]The number of filters $P$ should be large enough to characterize the scale space. We typically set $P = 7$.

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
