[Reviews · NeurIPS 2014]

Submitted by Assigned_Reviewer_26

The paper tackles the problem of blind image deconvolution and proposes an alternative approach to the commonly employed coarse-to-fine scheme in existing state-of-the-art methods. Instead of performing kernel estimation at different resolution scales, the authors suggest to apply Gaussian blur of various widths to yield different scales, each of which accentuates complementary image features. Kernel estimation at all scales is performed simultaneously rather than successively such that kernel information revealed in different scales can be combined in the estimation process. The weights that determine how much information each scale contributes to kernel estimation are chosen adaptively. Algorithmically the proposed approach follows closely [27]. In a number of examples the validity and robustness of the proposed method is demonstrated.

The paper is well written and clearly structured.

Some open questions and additional comments:
* What stopping criteria are used in Algorithm 1, 3. Please be more precise!
* Please provide more detailed information on the implementation! How does the proposed method scale with image size in terms of memory requirement and computational cost? What are typical runtimes?
* Equ. (3) doesn't follow from (2), since Gaussian filtering correlates neighboring pixels. Hence, $n_p$ is correlated and not iid Gaussian noise, which however underlies Equ. (3). Can you please comment on this?
* (S13) relies on the fact that $k$ and $f_p$ can be exchanged which is true for an invariant blur model but not for non-uniform blur. Can you please comment on this?
* Can you please be more specific how $h_{ip}$ is constructed with $w$ and the set of transformation operators in Section 3.3 using the generalized observation model of [19]? What is meant by "local kernel at site $i$"? If the fast forward model of (Hirsch et al., Fast removal of non-uniform camera shake, ICCV 2011) is used similar to [27], then this should be cited.
* "The deblurred image from the propose method..." -> "The deblurred image from the proposed method..."

Summary: Although the proposed algorithm is very similar to previous work [27], the paper provides an interesting view point to the commonly accepted and employed coarse-to-fine scheme in existing state-of-the-art methods. It seems likely that the paper will trigger follow-up work.

Submitted by Assigned_Reviewer_31

The paper starts from the hypothesis that image structures that are “small scale” or with gradient profiles distributed near-each other are invariably recovered incorrectly when the size of the blur kernel is similar to their size. A consequence of this is poor kernel estimates
intentionally blurring as done in a coarse-to-fine deblurring process naturally suppresses the contributions of these small scale structures since their contributions is downplayed at coarser levels.
Building on these ideas, the paper suggests a joint blink deblurring across the different scales with the constraint that the latent image remains the same for all scales.
Parameters are tuned/learnt to account for different levels of noise

Pros
- Novel idea of jointly deblurring across multiple scales with constraints.
- Improvement over competing methods.
Cons
- Claims of newer analysis not supported or overstated (?)
- Details on non-uniform blur models missing

1) The idea of jointly deblurring across multi-scales, to the best of my knowledge, is new.
it raises some question though.
First, wouldn’t such a method over penalize errors in the coarser scales — since they are present in all the scales ?
Second, the paper suggests the finer scales are less robust to noise and while coarser scales are robust to noise, they lack fine detail.
While i agree with this motivation, does the proposed method make better use of it than other techniques — say Fergus et al. where a coarse to fine refinement is performed under the same motivation.

2) The paper keeps suggesting that while similar techniques have been proposed, the paper provides motivation and analysis.
I am not sure what analysis the paper refers to.
There are experimental results that suggests that the joint multi-scale deblurring has benefits.

A lot of claims are made in lines 262-294. But these are largely empirical and many details are missing. What was the sample size with respect to number of latent images, number of blur kernels, and hence, wouldnt quite pass for a rigorous analysis.

3) The paper could do a better job in describing the non-uniform model since they are used in the real examples. Specifically, reporting how the homographs were generated and the number used etc. should help reproducibility of the results.

Smaller comments
- As opposed to “Estimation error” which is very hard to interpret, the paper should report scores with SNR or PSNR. There are normalized and in log-scale — both of which make it significantly easier to interpret.

- Line 283. “This corroborated previous findings” — citations are required here.
- Figure 6. Pl show recovered blur kernels for all images. These are valuable in interpreting the results.
Summary: The idea is novel and the results are promising. However, there need to be a more thorough study of the joint scale-space deblurring.

Submitted by Assigned_Reviewer_32

Multi-scale approaches are known to be a beneficial strategy in blind image deblurring. The authors propose a new variational formulation for image deblurring in the presence of noise by adding up regularized energy functionals for deblurring at different scales. In each of those functionals a scale filter, e.g., a Gaussian filter, is applied to the blurred and noisy given image and the (unknown) blur kernel.

Although multi-scale approaches are not new in image deblurring, the specific variational model proposed here seems to be novel. What could be improved is the statistical motivation and derivation of the model. Already in (3) it should be mentioned that this likelihood depends on the noise model, i.e., that this is tailored to Gaussian noise. For a reader who is unfamiliar with the topic, the step from (3) to (4) is hard to understand. It is easily overlooked that there is a sparsity prior involved here which does not become clear by just looking at p(x). Although this approach is not new and can be found in the papers cited here, one should make the paper more self-contained and explain this step in more detail since it is crucial for the obtaining the model (5). Similarly, it would be good to put (6) in a statistical context and motivate why one ends up with a sum over the scales and why the \lambda_p's appear as they do. This would also explain a bit better what learning the \lambda_p's means since it might be that this just downweighs the importance of small-scale structures just because it is cheaper to deblur a more strongly filtered image. In this context, one could also explain a bit more why the model does not need a coupling between the different parameters \lambda_p.
What is also missing a bit is the discussion of (6) from the point of view of optimization, i.e., one should discuss things like the existence of a solution, (non-)smoothness, (non-)convexity and how this effects the practical solution of the problem.

Finally, the authors might want to consider the following reference which is connected to their approach: "Inverse scale space methods for blind deconvolution" by A. Marquina

Remark: There a quite a few typos and grammatical errors in the text, e.g.:
- l. 32: a driven example
- l. 356: the the

Summary: The model proposed here combines existing approaches (i.e. the deblurring functional and the multi-scale technique) in a straightforward way and the numerical results look convincing. What is missing is a more rigorous statistical motivation and a mathematical analysis of the resulting variational model.
Author Feedback
Author rebuttal: To R1
*We will provide more implementation details as suggested. We use the number of iterations as the stopping criteria and terminate the algorithm after a fixed number of iterations, which is 15 in practice. The method scales linearly with kernel size and image size. Typical run time for estimating a 25*25 kernel from a 256*256 image is 2-3 minutes.

*Gaussian filtering will correlate neighbor pixels. But we are talking about noise here. If the original noise is iid Gaussian in (2), after Gaussian filtering it is still iid Gaussian.

*For non-uniform blur, the blur kernel at different image pixels can be different. However, for a small local region, the blur kernel can be approximated using a locally constant blur. In this case, $k$ and $f_p$ can be exchanged locally under the locally constant blur approximation.

*By "local kernel at site $i$", we refer to the blur kernel corresponding to the i-th image pixel induced by non-uniform camera shake and filter $f_p$, which is constructed as $h_{ip} = f_p*(\sum_i w_i P_i e_i)$. $e_i$ denotes an image with all zero elements except at the place corresponding to the i-th pixel. $P_i$ is the i-th projection operator. We will add citation to Hirsch et al.

To R2
*As we are working in the gradient domain, different scales will reveal different information about the spectrum. While coarser scales present in all the scales, the contributions of different scales are adaptively balanced by a set of weight parameters (\lambda_p) learned for the input image, alleviating the problem of over penalization.

*The conventional coarse-to-fine method such as Fergus et al. is mainly motivated to exploit the solution space effectively, by initializing the kernel estimation at a finer scale using the estimation from a coarser scale. The problem of the successive coarse-to-fine scheme is that in the presence of image noise, while the kernel estimation is typically reasonable at coarser scales, it can diverge at finer scales due to the increased impact of image noise. In addition to the property of coarse-to-fine estimation, the proposed approach is more effective than conventional coarse-to-fine methods in presence of noise due to the joint estimation scheme across scales, as shown in Fig.1 and Fig.5c, where the kernel estimation of conventional coarse-to-fine methods (Levin and Zhang) diverge, while the proposed method can produce a more reasonable estimation.

*We summarize in lines 262-294 several interesting observations with our algorithm in experiments. They are empirical observations rather than strong claims. We observed same properties across different images/blurs, i.e., all the test images in the main paper and the supplementary file.

*For the non-uniform model, the homographies are generated by point-sampling a motion space (translation and rotation), and the range of the space is set to cover the estimate of the kernel size. We have shown the recovered blur kernels in the supplementary file due to space limitation.

To R3
*Although multi-scale approaches are commonly used in blind image deblurring, the way we use multi-scale information is novel, and very effective in presence of noise. Compared to conventional multi-scale methods, the kernel estimation in the proposed method is performed at all scales simultaneously rather than successively. Therefore, kernel information revealed in different scales can be combined adaptively in the estimation process. This is especially beneficial when the input image is noisy. In the successive coarse-to-fine scheme, while the kernel estimation is reasonable at coarser scales, it can diverge at finer scales in presence of noise, as shown in Fig. 1. In contrast, our joint formulation is much more robust.
Furthermore, our formulation is not limited to Gaussian filters and can also be used with directional filters as in [28].

*The cost function (5) is derived using type-II estimation with Bayesian modeling, and (6) is then constructed as the summation of (5) across all the sales. While (6) constructed this way has a clear energy-minimization interpretation, it can also be interrelated probabilistically. From this perspective, the summation over scales can be obtained by assuming the likelihood functions (3) of different scales are conditionally independent and the joint likelihood function can be obtained via a product over scales, which turns into summation over scales when taking negative log of the joint distribution. A graphical model is provided in Fig.1c of the supplementary file. \lambda_p measures the fidelity and or cost of deblurring at the corresponding scale. While currently there is no coupling between different lambda_p parameters, certain types of coupling via smoothness/sparsity penalty can be applied to promote different types of parameter forms if desired.

*Thanks for the comment. We will make it clear that the likelihood (3) is tailored to Gaussian noise. The step from (3) to (4) was not elaborated due to space limitations. However, we agree with the reviewer and will provide more explanation of this step in the paper.

*We will add more discussion from the optimization perspective. The cost function in (6) is a non-convex one, thus there is no guarantee for global optimal. However the algorithm takes advantage of the majorization-minimization technique for principled non-convex optimization, which helps to avoid premature convergence to a bad local minima by progressively introducing more concavity (sparsity) of the penalty function automatically according to the noise level and blur estimation. Because of this, the algorithm does not sensitive to the initialization of blur parameter and noise level. In practice, we initialized the blur close to delta kernel and noise level to be a fixed value (0.05) over all the experiments (pixel value is in [0 1]).

*Thanks for the suggested reference. We will add citation to it.